# Fluconazole-Induced Protein Changes in Osteogenic and Immune Metabolic Pathways of Dental Pulp Mesenchymal Stem Cells of Osteopetrosis Patients

**DOI:** 10.3390/ijms241813841

**Published:** 2023-09-08

**Authors:** Zikra Alkhayal, Zakia Shinwari, Ameera Gaafar, Ayodele Alaiya

**Affiliations:** 1Therapeutics & Biomarker Discovery for Clinical Applications, Cell Therapy & Immunobiology Department, King Faisal Specialist Hospital & Research Centre, P.O. Box 3354, Riyadh 11211, Saudi Arabia; szakia@kfshrc.edu.sa (Z.S.); agaafar@kfshrc.edu.sa (A.G.); 2Department of Dentistry, King Faisal Specialist Hospital & Research Centre, P.O. Box 3354, Riyadh 11211, Saudi Arabia

**Keywords:** carbonic anhydrase II activators, human-exfoliated deciduous teeth, dental pulp, mesenchymal stem/stromal cells (MSSCs), proteomics, osteopetrosis, azole, fluconazole

## Abstract

Osteopetrosis is a rare inherited disease caused by osteoclast failure, resulting in increasing bone density in humans. Patients with osteopetrosis possess several dental and cranial complications. Since carbonic anhydrase II (CA-II) deficiency is a major cause of osteopetrosis, CA-II activators might be an attractive potential treatment option for osteopetrosis patients. We conducted comprehensive label-free quantitative proteomics analysis on Fluconazole-treated Dental Pulp Mesenchymal Stem/Stromal Cells from CA-II-Deficient Osteopetrosis Patients. We identified 251 distinct differentially expressed proteins between healthy subjects, as well as untreated and azole-treated derived cells from osteopetrosis patients. Twenty-six (26) of these proteins were closely associated with osteogenesis and osteopetrosis disease. Among them are ATP1A2, CPOX, Ap2 alpha, RAP1B and some members of the RAB protein family. Others include AnnexinA1, 5, PYGL, OSTF1 and PGAM4, all interacting with OSTM1 in the catalytic reactions of HCO3 and the Cl- channel via CAII regulation. In addition, the pro-inflammatory/osteoclast regulatory proteins RACK1, MTSE, STING1, S100A13, ECE1 and TRIM10 are involved. We have identified proteins involved in osteogenic and immune metabolic pathways, including ERK 1/2, phosphatase and ATPase, which opens the door for some CA activators to be used as an alternative drug therapy for osteopetrosis patients. These findings propose that fluconazole might be a potential treatment agent for CAII- deficient OP patients. Altogether, our findings provide a basis for further work to elucidate the clinical utility of azole, a CA activator, as a therapeutic for OP.

## 1. Introduction

Osteogenesis is a dynamic process in which the osteoclast and osteoblast play critical roles [1,2]. Osteoclasts play a key part in bone remodeling, which involves both bone resorption by osteoclasts and bone formation by osteoblasts. Osteoclasts were first described long ago as hematopoietic cells, when osteoclast function was restored in mi/mi mice after receiving hematopoietic cells from normal siblings, and osteopetrosis was produced in normal recipient mice after receiving hematopoietic cells from osteopetrotic mice with defective osteoclast function [3]. As monocytic lineage-derived cells, they belong to a family of cells that displays a wide heterogeneity and plasticity and that is involved in phagocytosis and innate immune responses [4]. At the same time, it was discovered that osteoclasts are much more than bone-resorbing cells. In particular, they were reported to play a role in bone immunology [5]. T cells were discovered to have an impact on osteoclast generation and activation, and these interactions are bidirectional. Modifications in their differentiation or bone-resorbing activity are associated with a number of pathologies ranging from osteopetrosis to osteoporosis, chronic inflammation and cancer, all of which are characterized by immunological alterations [5,6]. Alternatively, osteopetrosis (OP), or “marble bone disease”, is a group of rare heterogeneous hereditary disorders that are characterized by osteoclast resorption or differentiation failure, resulting in a high density of bone. This results in thick, sclerotic, and fragile bones. High bone density on radiographs is a common symptom of this condition, first recognized in 1904 by the German radiologist Albers-Schonberg [7]. This most noticeable sign of increased bone density on radiographs is caused by low osteoclast activity, which reduces bone resorption [8,9]. Instead of conferring strength, the overly dense bone architecture leads to brittleness, which predisposes to fracture and subsequent infections. Normal bone modeling and remodeling disruption can result in skeletal deformity and dental abnormalities, as well as interfere with mineral homeostasis. At least nine distinct forms exist, each with its own mode of transmission and spectrum of severity from asymptomatic to fatal, contributing to the extreme variability. Depending on the severity of symptoms and the family history of the disease, different subgroups of human osteopetrosis have been identified [7,9,10]. Mutations in genes such as *CAII*, *CLCN7*, *OSTM1*, *PLEKHM1*, *SNX10*, *TCIRG1*, *TNFSF11* and *TNFRSF11A* have been reported and all linked to the pathogenesis of osteopetrosis [11,12,13]. Autosomal recessive osteopetrosis (ARO) and autosomal dominant osteopetrosis (ADO) are the two primary osteopetrosis categorization categories [10,14]. Mutations in the CAII gene can cause either a lethal disease phenotype or a benign variant [9,15]. On the other hand, it has been established that the pulp of primary deciduous exfoliating teeth is a rich source of large variety of progenitor/stem cells, such as SHED-stem cells [16,17]. Genetic studies have linked CAII enzyme defects to a wide range of phenotypic disorders, including osteopetrosis. Although the etiology is not known, most mutations that affect human osteoclast formation likely result in fatalities during embryonic development [14,18]. Renal tubular acidosis and deficiency in Carbonic anhydrase II are pathognomonic features of osteopetrosis, while CAII plays a critical role in the Co2 and bicarbonate reactions. Most cases of osteopetrosis are caused by mutations in genes encoding enzymes involved in the acidic erosion of the bone matrix by osteoclasts in the lacuna between the bone matrix and the osteoclasts [14,18]. 

Analytical proteomics has shown promise as a method for discovering protein biomarkers for a variety of diseases, but only a small portion of this research has been translated into clinical practice. Recent work by our lab showed a global protein analysis of stromal cells and mesenchymal stem cells isolated from the dental pulps of patients with osteopetrosis who were also deficient in CAII [8]. Inhibitors of CA have been the subject of extensive research in the past and present due to the therapeutic promise they may hold in areas including cancer treatment, glaucoma prevention and osteoporosis and bone mineralization promotion. In contrast, research on the therapeutic potential of CA activators in the restoration of CAII deficiency has not been explored. In the early 1990s, the activation of the metalloenzyme carbonic anhydrase (CA, EC 4.2.1.1) was confirmed using highly purified enzymes, and precise techniques such as stopped-flow assay and X-ray crystallography demonstrated this, as well as a general mechanism of action [19,20,21,22]. Alternatively, it was discovered that the pKa values for a number of azoles, bisazolylmethanes and bisazolylethanes correlated with the activating power of carbonic anhydrase II. Compounds with pKa values between 6.5 and 8.0 showed strong activations. Fluconazole is an oral and parenteral triazole antifungal. Its safety and pharmacokinetics make it a good alternative treatment for many fungal and parasitic diseases. It has a long half-life, high water solubility, and 10 times the plasma concentration in skin. Its azole derivatives make it suitable as a CA activator for in vitro testing.

This study aimed to explore the in vitro treatment of DPMS-derived cells with exogenous CA activators to restore some CA II activity as possible therapeutic targets for osteopetrosis patients. We used label-free quantitative expression proteomics as an objective tool to investigate the pathophysiological changes induced by CA activators and to identify therapeutic targets of fluconazole as a potential treatment of osteopetrosis patients.

## 2. Results

Isolated mesenchymal stem/stromal cells from dental pulp (DP-MSSCs) of recently extracted deciduous teeth from osteopetrosis (OP) patients were treated with serial doses of fluconazole and compared with untreated OP cells and healthy controls (HCs) untreated cell lysates were analyzed by label-free quantitative proteome profiling. The clinical characteristics and CAII status of all patients and control subjects are summarized in Table 1. We identified over 1500 proteins. Four different concentrations (1 µM, 2.5 µM, 5 µM and 10 µM) were used for pair-wise analysis of each of the treated doses of OP compared with untreated OP samples in order to see all treatment-induced changes across wide dose range. Only 251 proteins were significantly differentially expressed and represented across all the four doses. The observed changes were considered significant using both *p* value < 0.05 and at least >two-fold change. The 251 dataset was then evaluated between the three sample groups of HC untreated, OP untreated and OP treated samples. We have used 10 µM in the figure as a symbolic representation of maximum dose used. The changes in the expression of these proteins are depicted in Figure 1 and the full list is shown in Appendix A. Altogether, 820 proteins form the union of all differentially expressed proteins across all the four treatment doses. The dataset was not subjected to further evaluation and is listed in Appendix A.

### 2.1. Molecular Functional Classification of the Treatment Induced Protein Changes

We then examined the molecular functional classification of the treatment-induced 251 differentially expressed proteins between the sample groups using the Panther gene ontology (GO) pathway analysis program (Panther V17.0). Many of these proteins were represented in different molecular functional connections to the treatment-induced changes. Among the prominent functional category hits are binding, catalytic activity and structural molecular activity. Others include transporter, molecular functional regulator and ATP-dependent activity. The fractions of molecules represented in 11 different molecular functions are annotated in Figure 2 Left Panel and their expression changes denoted in heat maps as presented in Figure 2 Right Panel.

We used the knowledgebase Ingenuity Pathway Analysis (IPA, Qiagen, MD, USA) to explore the protein–protein interactions and implications of some of the differentially expressed proteins in relation to their cellular and functional features. This consists of a multiplicity of datasets to query and generate computational pathways and networks of interacting large-scale genes and gene products. Interestingly, the 251 proteins that were significantly different in their expressions between the three sample groups were involved in multiple signaling networks. Notably, among the top diseases and functions relevant to this study are Connective Tissue Disorders, Developmental Disorders, RNA Post-Transcriptional Modification, Cellular Assembly and Organization, Cellular Compromise and Cellular Response to Therapeutics. Others include Energy Production, Nucleic Acid Metabolism, Small Molecule Biochemistry Hematological Disease, Hereditary Disorder, Organismal Injury and Abnormalities. Excitingly, one of the hubs of the generated networks is ERK 1/2, in communications with other osteogenesis-implicated molecules like ATPase, Annexin and Calceneurin, along with other proteins that are biologically pertinent to the pathogenesis of OP as well as the CA catalytic reaction process. The interactions of these proteins are as illustrated in Figure 3 and Figure 4.

### 2.2. Fluconazole-Induced Protein Changes Involving Osteogenesis

Additionally, we examined the individual roles of these 251 differentially expressed proteins for possible connections in molecular features associated with osteopetrosis. Interestingly, 26 of them were involved in osteogenesis-related pathways. The majority of the 26 proteins were implicated in osteoblast regulation, while the rest were associated with osteogenesis imperfecta disease and osteoclast regulation. Only three proteins were found that had never been linked to osteogenesis before. The details of these proteins are summarized in Table 2. Likewise, these proteins were associated with forward and backward actions of CAII in the reaction between carbon dioxide and water (CO_2_ + H_2_O) leading to carbonic acid (H_2_CO_3_) and bicarbonate (HCO_3_), respectively. The expression changes of some of these proteins and their associated biological processes and osteogenesis-related interactions with CAII are shown in Figure 4 and Figure 5. 

## 3. Discussion

Several inhibitors of the metalloenzyme carbonic anhydrase II have been the subject of extensive drug discovery research in the past because of their therapeutic potentials in areas including treatment of cancer, glaucoma prevention, osteoporosis and promotion of bone mineralization. However, in contrast, research on the therapeutic potential of amines such as CA activators, including histamine and others, has not been explored. The aim of this study is to learn more about the potential therapeutics of CAII activators by describing fluconazole-induced protein expression changes in DPMS cells derived from CAII-deficient osteopetrosis patients. In this study, we have attempted to demonstrate the proof of concept that CA might be activated by using a biomedical agent to reverse the progression of osteopetrosis caused by genetic deficiency of this enzyme with associated clinical phenotypes involving structural malformation. 

We employed global untargeted label-free quantitative expression proteomics as a biomarker discovery tool to investigate and better our understanding of the underlying pathophysiological alterations caused by deficient CAII, as well as the potential mechanistic action of fluconazole as a CA activator in the restoration of inadequate osteogenesis in osteopetrosis patients. We anticipate that fluconazole treatment will induce changes in either upstream and/or targeting its downstream signaling effectors of CAII catalytic actions despite its deficiency in these patients.

We previously reported that the dental pulp of OP patients contains a population of mesenchymal stem/stromal cells (MSSCs) capable of multilineage differentiation comparable to that of healthy subjects’ DP-MSSCs [8]. Our isolated DP-MSSCs were shown to be CD44-, CD73-, CD105- and CD90-positive by immunohistochemistry and flow cytometry analysis. The proliferation, differentiation and protein expression patterns of these cells obtained from HC subjects and OP patients showed approximately 94.3% similarity. Interestingly, the observed minor differences of 86 differentially expressed proteins were found to be related to osteopetrosis disease. This present study reveals that 251 of more than 1500 total identified proteins implicated in osteogenic and immunological metabolic pathways, such as ERK 1/2, phosphatase and ATPase, were markedly differentially expressed in healthy subjects, untreated and azole-treated cells from osteopetrosis patients. Of relevance to this study, only 8 of the 86 differentially expressed proteins between undifferentiated healthy control and OP in our first reported paper were among the 251-protein dataset between fluconazole-treated osteopetrosis and untreated cells, as well as untreated healthy control cells. Two of these eight, ATPase Na+/K+ transporting subunit beta 3 (ATP1B3) and Protein Tyrosine Phosphatase non-receptor type 2 (PTPN12), were closely associated with CAII and were also represented among the interacting proteins in Figure 3’s network analysis. Interestingly, ATP1B3 is implicated in the upstream and downstream catalytic reactions of CAII. The two molecules were both significantly more upregulated with fluconazole treatment than previously reported [8]. PTPN12 is a non-receptor-type tyrosine-specific phosphatase that dephosphorylates both receptor and non-receptor protein tyrosine kinases. It also known as a negative regulator of various signaling pathways and biological processes such as hematopoiesis, inflammatory response, cell proliferation and differentiation and glucose homeostasis [23].

ATP1B3, an essential membrane protein, plays a crucial role in forming and upholding the Na and K ions’ electrochemical equilibrium across the plasma membrane [24]. The homeostatic gradient is essential for ionic regulation and the efficient transport of vital molecules, coupled with the exchange of Na (+) and K (+) ions across the plasma membrane for maintenance of urinary acidification. In our previous paper, PTPN12 was 2.4-fold more highly expressed in undifferentiated HC than OP, but in this study, it was 1215 times more downregulated in OP following fluconazole treatment. On the other hand, ATP1B3 was 5.1-fold more highly expressed in OP than HC-undifferentiated cells, and following fluconazole treatment-induced changes it was 18.3 times more overexpressed in OP than HC. One of the two proteins, ATP1B3, is implicated in interactions with other proteins in both upstream and downstream signaling cascades of CAII catalytic actions influencing osteoblast and osteoclast regulations, as illustrated in Figure 5.

It has been reported that osteoclasts play a role in bone immunology [5]. Changes in their differentiation or bone resorbing activity are linked to a variety of disorders, including immunopathologies. Furthermore, osteopetrosis-related bone brittleness increases the risk of fractures and accompanying infections, triggering immune responses. 

Its incapability to resorb bone has been linked to immune disorders such as osteopetrosis, osteoporosis, chronic inflammation and cancer [5,6]. T-lymphocytes have been linked to osteoclast development and activation. Consequently, more research is needed to uncover the function of the specific proteins identified in this study in the maintenance of a favorable immune response that might aid in the correction of osteoclast function following cell treatment by an activating agent, while it is well known that certain pharmacological agents that have been designed and approved for one particular medical use are sometimes discovered to be useful/effective for other problems [25]. Some of these pharmacological agents have been demonstrated to exert their effect, or rather some of their side effects manifest as pronounced CA-activating effects. Such actions include the phosphodiesterase IV inhibitor sildenafil, the b-blocker propranolol and members of the psychoactive compounds of the amphetamine and methamphetamine drugs [26,27,28]. 

Over the years, the genetic deficiencies of numerous CA isoforms (CA I, II, IV, VA, XII and XIV) have been described with associated clinical phenotypes such as osteopetrosis, retinal light response defect, pediatrics hyperammonemia, hyperchlorhidrosis and marble brain disease [29,30,31,32,33,34]. While some of these conditions might potentially be treatable using any of their corresponding activators, surprisingly, to our knowledge, none of the known CAAs have been employed on clinical samples as reported in this present study. While it has been demonstrated that activators for CA, particularly CA-II, might be beneficial for the treatment of bone deficiency diseases using the human osteogenic SaOS-2 cells in vitro stimulation by Ca-bicarbonate [35,36], the majority of results derived from in vitro studies are not usually translated into clinical settings. 

CA-II is a ubiquitous enzyme with dual roles in osteogenesis: inhibition of bone resorption on the one hand [37] and promotion of bone formation [38]. Therefore, maintaining this equilibrium is crucial for the catalytic action of CAII. It has been documented that Histamine64 (His64) is vital for the ionic interaction between the activator and carbonic anhydrase for its proton-shuttling function [39]. This docking site of the histamine, with the active part of carbonic anhydrase II facilitating the catalytic reaction with functional molecules and water [40], is highlighted in Figure 6. We have used fluconazole over histamine as the activator because of its ease of availability in the pharmacy. Like other azoles, it has a free nitrogen atom located on the azole ring capable of binding with the CAII docking site. Its pharmacokinetic properties by the intravenous (IV) and oral (PO) routes are highly comparable. Furthermore, its bioavailability following oral administration is greater than 90%, thus making it convenient as an oral antifungal agent [41]. Therefore, its azole derivatives make it suitable as a CA activator for in vitro testing.

In this study, we discovered that fluconazole used as a CA activator of cells derived from osteopetrosis patients induced significant protein changes, some of which are known to be associated with osteogenesis (Figure 5). Among the other identified proteins that are at least more than two-fold more highly expressed in treated cells were ATP1A2 and RAP1A, while CPOX, Ap2 alpha and some members of the RAB protein family were significantly downregulated in treated cells compared to untreated cells. All of these proteins regulate osteoblasts/osteocytes through the ERK1/2 osteogenesis pathway [43,44]. In addition, AnnexinA1 and AnnexinA 5 were 12- and 4.7-fold more overexpressed, respectively, following fluconazole treatment of OP cells, but PYGL and RACK1 were 33- and 4.1-fold more downregulated, respectively, by fluconazole treatment. These proteins interact with OSTM1 to facilitate the upstream enzymatic reaction of CAII in the catalysis of HCO3 and Cl- channel reaction [11,45,46,47]. Other identified proteins in this study that are involved in the catalytic reaction of CAII in connection with Ca2ATPase and Calcirumin A are ATP2A, ATP2B3, ATP1A2 and ATP1B3, and are between 2.7- and 18.3-fold more overexpressed in OP-treated than -untreated cells (details in Appendix A). In addition, other proteins were identified to be involved in the pro-inflammatory cytokine/VEGF pathway of osteoclast regulation. Among them, PPIF, MIF1, ECE1 S100A13 and RACK1 were all highly downregulated by azole treatment, while STING1 and TRIM10 were upregulated in treated cells. Overall, the protein–protein interactions and their involvement in the various stages of CAII catalytic actions induced by fluconazole treatment of OP cells further elucidate the disease-specific effect, making this drug a potential treatment option for OP patients (Figure 5). Fluconazole, a member of the azole family, has been shown to block V-ATPase in osteopetrotic patients [48]. Therefore, fluconazole may be administered to individuals with osteopetrosis as an antifungal preventative [49], while azole, as a V-ATPase inhibitor, has not yet been well studied in clinical trials, thus restricting its clinical application in V-ATPase targeting in osteopetrosis [19,20,21,22]. Therefore, an initial approach might be to evaluate both downstream and down-upstream changes by targeting specific isoforms and V-ATPases with resulting implications in osteopetrosis pathogenesis. We identified several ATPases, including ATP2A2, ATP2B3, ATP1A2 and ATP1B3, that play critical roles in signaling cascades that promote osteoclast differentiation and activation. All the above-listed identified ATPases were significantly upregulated in fluconazole-treated compared to -untreated cells.

Western blotting is a commonly used figurative validation method for quantitative proteomics data [50]. The lack of specific antibodies for some of the identified proteins of interest, particularly the highly abundant proteins, is one of the inherent limitations of validation by Western blot analysis. We have used some of our experimental approaches for both discovery and validation prior to using other validation methods like Western blotting. The fact that these proteins are linked to the known pathogenesis of osteopetrosis (specifically the ERK1/2/Annexin/NaK ATPase/Calcineurin pathway) in the Ingenuity Pathway Analysis (IPA) indirectly confirms that the biological information of these protein panels is related and relevant to changes induced by OP disease treatment agents (Figure 3 and Figure 5). We believe that the treatment agent will induce coordinated changes in the expression of these specific proteins involved. Additional in vivo validation is required to support the clinical utility of fluconazole treatment for osteopetrosis patients.

## 4. Materials and Methods

### 4.1. Patient Samples

The mesenchymal stromal cells were derived from the pulp of primary incisor and canine teeth from HC and OP subjects aged 6–15 years as previously described [8].

All patients were from one center, at the dental clinic of the King Faisal Specialist Hospital and Research Center. Details of inclusion and exclusion criteria were as previously described [8]. Briefly, all patients were confirmed to have a diagnosis of autosomal recessive OP with renal tubular acidosis due to carbonic anhydrase II deficiency. All healthy controls were subjects seeking dental evaluation and treatment at the hospital. Deciduous teeth comprising central and lateral incisors and canines were extracted from five (5) OP and six (6) healthy control subjects. The clinical characteristics of all patients and control subjects are summarized in Table 1.

The Research Advisory Council (RAC)/Office of Research Affairs (ORA), KFSH&RC approved the study (12-BIO 2343-20) and the protocol was conducted in compliance with the Declaration of Helsinki.

### 4.2. Cell Isolation, Culture and Fluconazole Treatment of Dental Pulp Mesenchymal Cells

The mesenchymal stromal cells were isolated from the dental pulp of recently extracted deciduous teeth from osteopetrosis (OP) patients as previously described [8]. Briefly, the exfoliated teeth were mechanically extracted and rinsed with 0.12% chlorhexidine gluconate and subsequently washed with a buffer containing PBS, 1% gentamicin and 1% fungizone. Each tooth was methodically cracked open using a low dental hand piece and the pulp was carefully exposed under aseptic conditions.

Following digestion, the sample was centrifuged at 1500 rpm for 5 min at 4 °C, and the pellet was re-suspended in fresh growth medium (α-MEM without phenol) (Gibco, Waltham, MA, USA) with 20% MSC-FBS (Gibco, Waltham, MA, USA), 1% non-essential amino acids (Sigma, St. Louis, MO, USA), 1% AA (Sigma, St. Louis, MO, USA), 1% ITS-A (Gibco), 2 µM L-glutamine (Sigma, St. Louis, MO, USA), 100 µM L-ascorbic acid 2-phosphate (Stem Cell Technologies, Vancouver, BC, Canada), 100 ng/mL EGF (Gibco, Waltham, MA, USA) and 40 ng/mL FGF (Gibco, Waltham, MA, USA). The cells were then placed in a 6-well Petri dish pre-coated with 10 µg fibronectin (Corning, NY, USA) for at least 1 h. at 37 °C, supplemented with culture medium in an incubator at 37° with 5% CO_2_. The medium was regularly changed after 2–3 days until 80–90% confluence. Observation under the microscope showed that more than 90% of the cells were viable and healthy prior to processing for proteomics analysis by mass spectrometry.

The mesenchymal stromal cells were treated for 24 h with 1, 2.5, 5 and 10 µM fluconazole (Pfizer Holding), Fareva Amboise, Pocé-sur-Cisse, France), and control untreated OP cells were successively trypsinized and collected for subsequent downstream proteomics analyses. Owing to the low throughput of LC/MS/MS-based quantitative analysis, we pooled an equal amount of total complex protein mixture from each of the treated and untreated osteopetrosis patients, as well as healthy control subjects, as detailed in our previous paper. The pools of each sample group were run at least twice or three times.

Below is the illustrated workflow of cell culture and downstream proteomics analysis work (Figure 7).

### 4.3. Whole-Cell Lysate of Mesenchymal Stromal Cells

Whole-cell lysates of the mesenchymal stromal cells treated with fluconazole and the control untreated cells were accomplished as previously described with minor modification [51,52]. Briefly, trypsinized cells were washed twice in phosphate-buffered saline containing protease inhibitor cocktails (Benzamidine and PMSF) and then centrifuged at 800× *g* and 4 °C for 3 min. Thereafter, the cells were centrifuged for 5 min at 2700× *g* (4 °C). The collected cell pellets were lysed in a solution containing 0.1% RapiGest SF (Waters, Manchester, UK) supplemented with protease inhibitors and total protein contents were determined using the Bradford method [53].

### 4.4. In-Solution Digestion of Complex Protein Mixture

A total of 100 μg of complex protein mixture from isolated mesenchymal stromal cell from was taken from the fluconazole-treated and control untreated OP mesenchymal stromal cells and subjected to in-solution protein digestion prior to LC–MS/MS analysis as previously described [54,55]. Briefly, protein denaturation was achieved in 0.1% RapiGest SF (Waters Corp., Manchester, UK) at 80 °C for 15 min, reduced in 10 mM DTT at 60 °C for 30 min and then alkylated in 10 mM Iodoacetamide (IAA) for 40 min at room temperature in the dark. All samples were trypsin-digested (*w*/*w*; 1 μg/μL trypsin concentration, (Promega, Madison, WI, USA)) at 1:50 enzyme/protein ratio at 37 °C overnight and the reaction was quenched with 12M HCL. All samples were diluted with aqueous 0.1% formic acid to protein concentrations of 1 μg/μL at the end of digestion and spiked with yeast alcohol dehydrogenase (ADH; P00330) as an internal standard for absolute quantitation prior to LC/MS analysis as previously described [8,54,55].

### 4.5. Protein Identification by Label-Free Liquid Chromatography Mass Spectrometry (LC/MSE) 

Samples were subjected to label-free quantitative expression proteomics using 1-dimensional Nano Acquity liquid chromatography coupled with mass spectrometry on a Synapt G2 HDMS instrument (Waters Scientific, Manchester, UK). The Electrospray Ionization (ESI) Mass Spectrometry analysis was optimized and all acquisitions were carried out on Trizaic Nano source (Waters Scientific, Manchester, UK) ionization in the positive ion mode nanoESI as previously described [8,56,57]. An equal amount of protein digest (3 μg) was loaded on column (Acquity Trizaic HSS T3 Nano tile, Waters) and samples were processed using the Acquity sample manager with mobile phase comprising of A1 (99% water/1% Acetonitrile/0.1% formic acid) and B1 (100% Acetonitrile + 0.1% formic acid) with a sample flow rate of 0.450 μL/min. All data acquisition was performed using iron mobility separation experiments (HDMSE) over a range of *m*/*z* 50–2000 Da with a scan time of 0.9 s, incremental transfer collision energy 20–50 V with a total acquisition time of 120 min. Data were acquired using the Mass Lynx programs (version 4.1, SCN833; Waters) operated in resolution and positive polarity modes, and each sample was analyzed in triplicate runs (as a measure of reproducibility). All automated data processing and database searching was performed using Progenesis QI for proteomics (Progenesis QIfp version 2.0.5387) (Nonlinear Dynamics/Waters). The generated peptide masses were searched against the specie–specific protein sequence Uniprot database (www.uniprot.org, accessed on 22 December 2022).

### 4.6. Data Analysis and Informatics

Quantitative Trans Omics Informatics (Waters) was used to process and search the generated data using Uniprot species-specific (human) reviewed entries. The following criteria were used for the search: one missed cleavage, maximum protein mass 1000 kDa, trypsin, carbamidomethyl C fixed and oxidation M variable modifications. Normalized label-free quantification was achieved using the Progenesis QIF proteomics software (QI V.4.0) for proteomics (Nonlinear Dynamics, Newcastle/Waters, UK). The data were filtered to show only statistically (ANOVA) significantly altered proteins (*p* ≤ 0.05) with ≥3 peptides identified and a fold change of more than 1.5. Additionally, ‘Hi3′ absolute quantification was performed using ADH as an internal standard to give an absolute amount of each identified protein (Waters).

## 5. Conclusions

We have identified proteins that are implicated in osteogenic and immune metabolic pathways, including ERK 1/2., phosphatase and ATPase, some of which were involved in CA activity. These protein changes were induced in fluconazole-treated MSCs from osteopetrosis patients, thus opening the door for the use of some of the CA activators as alternative drug therapies for osteopetrosis patients once fully validated. Altogether, our findings provide a basis for further in vivo work supporting that fluconazole, a CA activator, might be a potential therapeutic agent for CAII-deficient OP patients.

## Figures and Tables

**Figure 1 ijms-24-13841-f001:**
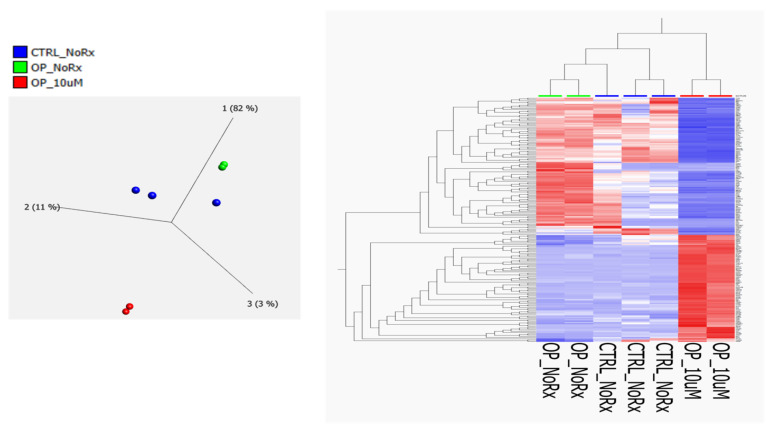
(**Left Panel**) Principal component analysis using the expression dataset of 251 significantly differentially expressed proteins between fluconazole-treated and untreated OP cells, as well as untreated healthy control cells. OP_NoRx = osteopetrosis untreated, OP_10 μM, osteopetrosis treated with fluconazole. (**Right Panel**) Hierarchical cluster analysis using the expression dataset of the 251 differentially expressed proteins as described above. The heat map depicts the differences in protein expression between the three sample groups. Image was created using the Qlucore Omics Explorer version 3.7, (Lund, Sweden) (https://qlucore.com, accessed on 29 June 2023).

**Figure 2 ijms-24-13841-f002:**
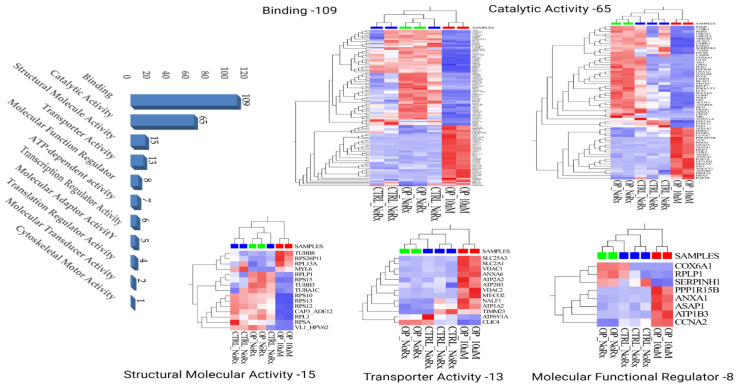
(**Left Histogram Panel**) Panther GO molecular functional classification of the treatment-induced 251 differentially expressed proteins. (**Right Panel**) Heat maps of some of the different molecular functional representations highlighting the expression changes of the implicated proteins between the treated and control sample groups. OP_NoRx = osteopetrosis untreated, OP_10 μM, osteopetrosis treated with fluconazole [the enrichment scores were significant for the listed cellular processes, with *p* < 0.05]. The heat map mages were created using the Qlucore Omics Explorer version 3.7, (Lund, Sweden) (https://qlucore.com, accessed on 29 June 2023) and panels assembled using Bio Render Scientific Image and Illustration Software, www.biorender.com, accessed on 29 June 2023.

**Figure 3 ijms-24-13841-f003:**
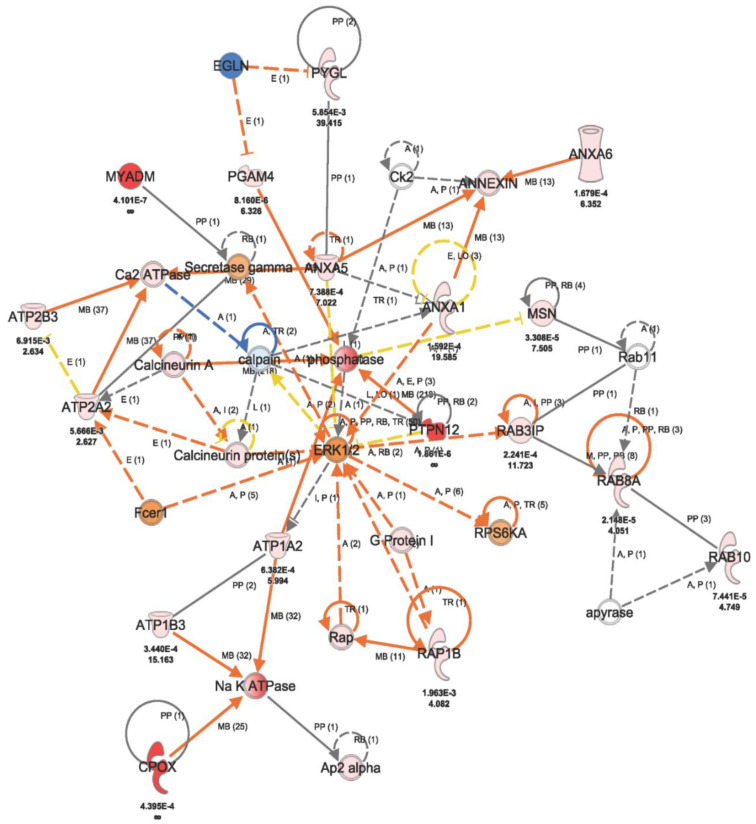
Ingenuity pathway analysis (IPA) showing protein–protein interactions of some of the 251 differentially expressed molecules between osteopetrosis cells treated with fluconazole and untreated cells as well as healthy control cells. Highlighted in pink or red colors are some of the implicated proteins from this study, with their expression fold-changes and *p* values indicated below each protein. Central to the network is ERK 1/2, with direct (solid lines) and indirect (broken lines) interactions with other osteogenesis-implicated molecules like ATPase, Annexin and Calceneurin. Other proteins without expression values below them represent molecules from the IPA database that altogether make up the non-canonical network. The network analysis was generated using the IPA program. (QIAGEN Inc., https://www.qiagenbioinformatics.com, accessed on 29 June 2023).

**Figure 4 ijms-24-13841-f004:**
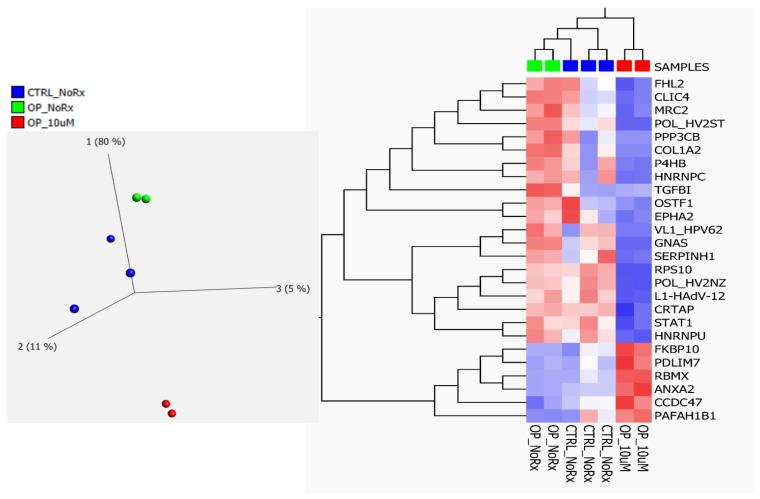
(**Left Panel**) Principal component analysis using the expression dataset of 26 out of 251 significantly differentially expressed proteins between fluconazole treated and untreated OP cells, as well as untreated healthy control cells, that were implicated in the osteogenesis pathway. (**Right Panel**) Hierarchical cluster analysis with the heat map showing the expression changes of the 26-protein dataset among the sample groups. (OP_NoRx = osteopetrosis untreated; OP_10 μM, osteopetrosis treated with fluconazole).

**Figure 5 ijms-24-13841-f005:**
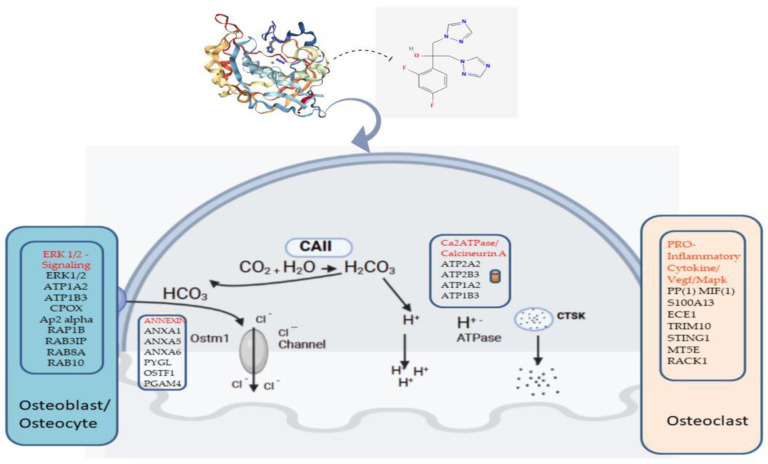
Overview of identified fluconazole-induced protein changes and their associated upstream and downstream biological processes and osteogenesis-related pathways. Listed below each panel are some of the currently known genes that are involved in osteogenesis pathways and listed in each panel are the identified proteins induced by fluconazole. Some of the proteins interact with ERK 1/2 signaling, Annexin, Ca2ATPase and pro-inflammatory molecules as depicted in Figure 3. (Image created using BioRender Scientific Image and Illustration Software, www.biorender.com, accessed 29 June 2023).

**Figure 6 ijms-24-13841-f006:**
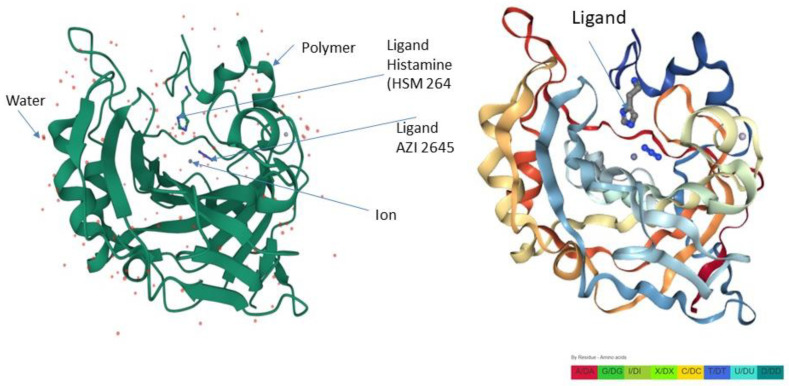
(**Left Panel**) Representative structures of the human carbonic anhydrase II highlighting the interacting docking site between the activator (histamine) and carbonic anhydrase. The ionic interaction with other functional water molecules between the activator and carbonic anhydrase facilitates the catalytic reaction. (**Right Panel**) Same 3-D skeletal structure with residue amino acid color. Image was created with the PDB ID and associated publication, NGL Viewer (AS Rose et al. (2018) NGL viewer: web-based molecular graphics for large complexes [42].

**Figure 7 ijms-24-13841-f007:**
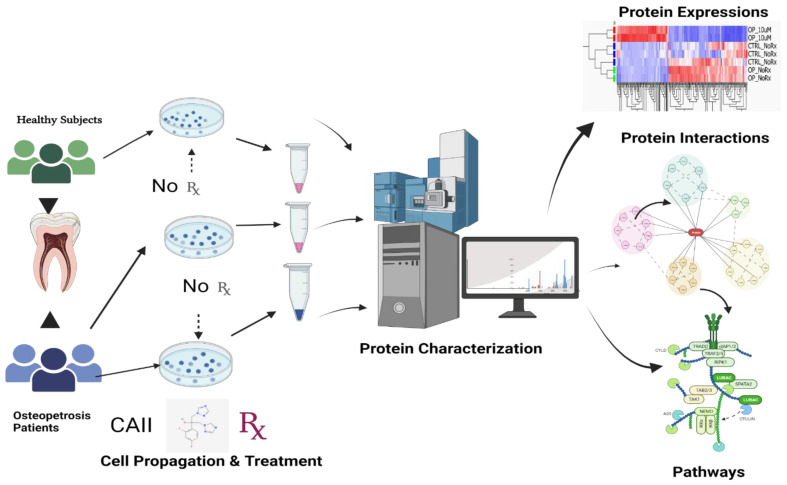
Workflow illustration: representative mesenchymal cells were isolated from the dental pulp of recently extracted deciduous teeth from osteopetrosis (OP) patients and healthy control (HC) subjects as previously described [8]. The extracted cells were treated with fluconazole. Both treated OP and non-treated OP cells, as well as healthy control subjects, were characterized by their protein expression changes using label-free quantitative liquid chromatography–tandem mass spectrometry (LC–MS/MS) for protein–protein interaction and implicated pathways. Image created using Bio Render Scientific Image and Illustration Software, www.biorender.com, accessed on 29 June 2023.

**Table 1 ijms-24-13841-t001:** Clinical and pathological characteristics of study patients.

Subjects	Age (Years)	Gender	Pathogenetics	Skeletal Radiology	Clinical Findings
OP#1	11	M	c.232 + 1G > A, IVS2 + 1G > Aof the CA2 gene, (AR, RTA)	High frequency of fractures, dense bones	Dental abnormalities, developmental delay, optic atrophy/esotropia, ADHD
OP#2	10	F	c.232 + 1G > A, IVS2 + 1G > Aof the CA2 gene (AR, RTA)	Dense skull bones	Short stature, esotropia and optic atrophy
OP#3	16	F	CAII deficiency Renal Profile Low CO2, (RTA)	Diffuse increased density of skeletal bones, sclerosis more prominent in skull	ADHD, developmental delay learning disabilities, scoliosis
OP#4	15	M	CAII deficiency Renal profile Low CO2, (RTA)	Brachiocephaly, increased bone density, Erlenmeyer flask deformity	Failure to thrive, fractures, intracranial calcification
OP#5	9	M	c.232 + 1G > A;IVS2 + 1G > A (RTA), AD, CAII	Generalized increase in bone density shape normal	Calcification, fractures
HC#1	11	F	Healthy	Not indicated	No Abnormalities
HC#2	9	F	Healthy	Not indicated	No Abnormalities
HC#3	10	M	Healthy	Not indicated	No Abnormalities
HC#4	16	M	Healthy	Not indicated	No Abnormalities
HC#5	9	M	Healthy	Not indicated	No Abnormalities
HC#6	13	F	Healthy	Not indicated	No Abnormalities

Footnote: CA-II, carbonic anhydrase II; RTA, renal tubular acidosis; ADHD, attention deficit hyperactivity disorder; AR, autosomal recessive; AD, autosomal dominant; IVS2, Intron 2.

**Table 2 ijms-24-13841-t002:** Characterization of some of the identified proteins that differed significantly following fluconazole treatments of cells derived from osteopetrosis patients. Details of the 251 differentially expressed molecules between osteopetrosis fluconazole-treated and -untreated cells, as well as healthy control cells, are described in Appendix A.

Accession	GN	Anova (p)	Max Fold Change	Highest Mean Condition	Lowest Mean Condition	Role in Bone Related Disease Association
O75718	CRTAP	0.0193	3.30	OP-NoRx	OP-Rx	Ostegenesis Imperfecta
P05962	POL_HV2NZ	0.0001	81.79	OP-NoRx	OP-Rx	Not known
P07237	P4HB	0.0002	3.32	OP-NoRx	OP-Rx	Ostegenesis Imperfecta
P07910	HNRNPC	0.0008	3.48	OP-NoRx	OP-Rx	Osteoblast
P08123	COL1A2	0.0305	170.43	OP-NoRx	OP-Rx	Ostegenesis Imperfecta
P16298	PPP3CB	0.0010	42.88	OP-NoRx	OP-Rx	Osteoclast
P20876	POL_HV2ST	0.0000	22.26	OP-NoRx	OP-Rx	Not known
P29317	EPHA2	0.0444	7.03	OP-NoRx	OP-Rx	Osteoblast-Clast
P36712	L1-HAdV-12	0.0004	5.12	OP-NoRx	OP-Rx	Not known
P42224	STAT1	0.0461	2.29	OP-NoRx	OP-Rx	Osteoclast
P46783	RPS10	0.0041	1850.80	OP-NoRx	OP-Rx	Osteoblast-Clast
P50454	SERPINH1	0.0000	5.35	OP-NoRx	OP-Rx	Oste Imperfecta
P50823	VL1_HPV62	0.0000	∞	OP-NoRx	OP-Rx	Not known
Q5JWF2	GNAS	0.0015	42.22	OP-NoRx	OP-Rx	Osteoblast-Clast
Q9UBG0	MRC2	0.0151	8.64	OP-NoRx	OP-Rx	Osteoblast
Q9Y696	CLIC4	0.0007	9.14	OP-NoRx	OP-Rx	Osteoblast
Q00839	HNRNPU	0.0117	2.46	OP-NoRx	OP-Rx	Osteoblast
Q14192	FHL2	0.0834	8.20	OP-NoRx	OP-Rx	Osteoblast
Q92882	OSTF1	0.0109	4.56	OP-NoRx	OP-Rx	Osteoblast
Q15582	TGFBI	0.0009	4.02	OP-NoRx	OP-Rx	Osteoblast
P07355	ANXA2	0.0000	6.68	OP-Rx	OP-NoRx	Osteoblast
Q9NR12	PDLIM7	0.0004	4.86	OP-Rx	OP-NoRx	Osteoblast
Q96A33	CCDC47	0.0014	3.43	OP-Rx	OP-NoRx	Osteoblast
Q96AY3	FKBP10	0.0030	5.92	OP-Rx	OP-NoRx	Oste Imperfata
P38159	RBMX	0.0020	979.35	OP-Rx	OP-NoRx	Osteoblast
P43034	PAFAH1B1	0.0085	39.22	OP-Rx	OP-NoRx	Osteoclast

## Data Availability

All data generated or analyzed during this study are included in this published article and its Appendix A. All methods are as detailed in this manuscript. The accession number for the protein identification and characterization data reported in this paper is based on Uniprot format. The raw mass spectrometry data generated in this study using Waters Synapt G2 HDMSE are available from the corresponding authors on reasonable request.

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
