# Peer review of "Fluconazole-Induced Protein Changes in Osteogenic and Immune Metabolic Pathways of Dental Pulp Mesenchymal Stem Cells of Osteopetrosis Patients"

_ijms, 2023, doi:10.3390/ijms241813841_

Round 1

Reviewer 1 Report

comments and questions are in the attached file

overall, quality of English is good, occasionally, spelling errors and missing words occur

Author Response

ID: ijms-2500580

We thank the reviewers and the Editorial Office for their critical review of our manuscript. We are pleased to inform you that the manuscript has been thoroughly revised and all the raised concerns were responded to adequately. Please find below our point-by-point responses to all the raised questions/ suggestions.

Reviewer 1

Comments and questions: -

  1. Reviewers’ comment: Fluconazole is not properly described and referenced in the article. Its action as a CAII activator is not documented. Is it a specific activator of CAII or can it also activate other CA isoforms? If yes, which carbonic anhydrases are expressed in MSC samples and what would be the consequences of their possible activation in these settings? Figure 6 shows Ligand Histamine and Ligand AZI but no explanation is given.

Response: In the early 1990s the activation of the metalloenzyme carbonic anhydrase (CA, EC 4.2.1.1) was confirmed using highly purified enzymes, and precise techniques such as stopped-flow assay and X-ray crystallography demonstrated this, as well as a general mechanism of action (Nabih Lolak, et al 2018; Suleyman Akocak et al 2019). Alternatively, it was discovered that the pKa values for a number of azoles, bisazolylmethanes, and bisazolylethanes correlated with the activating power of carbonic anhydrase II. Compounds with pKa values between 6.5 and 8.0 showed strong activations (Claudiu T. SUPURAN, 1993). Fluconazole, an oral and parenteral triazole antifungal. Its safety and pharmacokinetics make it a good alternative treatment for many fungal and parasitic diseases. It has a long half-life, high water solubility, and 10 times plasma concentration in skin. It is important to note that the specific role of CA in MSC biology and therapeutic application is still being researched. This has now been stated in the revised manuscript.

  1. Reviewers’ comment: In your recent review (Alkhayal, Z.; Shinwari, Z.; Gaafar, A.; Alaiya, A. Carbonic Anhydrase II Activators in Osteopetrosis Treatment: A Review. Curr. Issues Mol. Biol. 2023, 45,1373–1386) you state that fluconazole „has been shown to block V-ATPase in osteopetrotic patients“, although, offtarget blocking of V-ATPase could be harmful and the review states the necessity of specific inhibition. This fact was also not mentioned and discussed in the article.

Fluconazole, a member of the azole family, has been shown to block V-ATPase in osteopetrotic patients. Therefore, fluconazole may be administered to individuals with osteopetrosis as an antifungal preventative [56]. While azole as a V-ATPase inhibitor has not yet been well studied in clinical trials thus restricting its clinical application in V-ATPase targeting in osteopetrosis [57]. Therefore, initial approach might be to evaluate both down steam and down upstream changes by targeting specific isoforms and V-ATPases with resulting implications to osteopetrosis pathogenesis. We identified several ATPases including ATP2A2, ATP2B3, ATP1A2 and ATP1B3 that are playing critical roles in signaling cascades that promote osteoclast differentiation and activation. All the above listed identified ATPases were significantly up regulated in fluconazole treated than untreated cells.

  1. Table 1 with clinical characteristics is not included in the article. In the Results you report data for 3 control healthy samples, 2 untreated OP patients and 2 treated OP patients. You mentioned 6 healthy control subjects in M&M. How did you select 3 control healthy samples used in the analysis? According to hierarchical clustering (Fig 1, 2) one Ctrl healthy sample is more associated with OP untreated samples than with healthy controls, please comment.

Response: We inadvertently omitted the statement to state that all samples used in this study were as detailed in table 1 of our previous paper. However, Table 1 has now been added.

Owing to the low throughput of LC/MS/MS based quantitative analysis, we have pooled equal amount of total complex protein mixture from each of the patients as pooled of treated and untreated osteopetrosis patients as well as healthy control subjects as detailed in our previous paper. We have now clarified this under materials and methods. The pooled healthy control samples was run in triplicates while that of OP treated and un-treated were of duplicate runs as detailed under materials and methods section. The unsupervised hierarchical cluster analysis with expression patterns represented in the heat map is a reflection of the relatedness of the samples. It is clear from the dendrogram that the samples have separate lower nodes, but flipping the upper nodes will bring the OP untreated closer to OP treated.  However, it show in all the dendrogram the clear similarities between the two untreated and fluconazole treated sample indicating treatment induced changes.

  1. You report that all OP patients have CA II deficiency. Is it caused by mutation (I presume these data will be given in Table 1)? Is mutated CA II enzymatically active? Can mutated CA II be activated by fluconazole? Are there differences between CA II levels in OP samples and healthy samples?

Response: Fundamentally, gene mutation is a result of change or rearrangement in the DNA base sequence, owing to alterations in the resulting synthesized amino acids. This alteration often results in defective functional property of protein or enzyme. Osteopetrosis results from deficient CAII and even though a mutated CAII might not be active, we anticipate that treatment with CAII activator will induce changes in either upstream and/or targeting its downstream signaling effectors. The beauty of this paper using global untargeted proteomics approach is that our results shows changes induced following fluconazole identified both up-steam as well as downstream signaling canonical/non canonical protein changes. Related to CAII. We did not identify CAII in both OP and HC samples probably due to the detection limit of our analysis method as copy number of a molecule are proportional to their detection limits based on different analysis platforms such as antibody-based or MS based. This has now been stated in the revised manuscript.

  1. - How relevant are pathway changes (Figure 2) – please provide enrichment score or some other indicator? Are the same pathways changed after fluconazole treatment of OP samples and when compared OP and healthy samples?

Response: The 251 proteins were those that were differentially expressed significantly between fluconazole treated and untreated cells. The changes were significant using both p value < 0.05 and at least > 2-fold change. The enrichment scores were significant for the listed cellular processes, with p < 0.05; however, we did not include the enrichment scores so as not to over crowd the figure1.

  1. - Heatmaps (Figures 2, 3, 4) and hierarchical clustering show that fluconazole treatment led to considerable differences also when compared with normal healthy samples. Could these changes have some adverse effects?

Response: The answer to that question is beyond the scope of this study. However, validation of these findings might require in vivo functional studies model before fluconazole could be recommended for treatment of osteopetrosis patient. Furthermore, the recommended dose of fluconazole currently being used as anti-fungal treatment of oral fungal infection is safe. Therefore, we belief that there has not been any reported cases of untoward effect among osteopetrosis patient that have been treated for oral infection with antifungal agents. Therefore we belief that recommended dose of fluconazole should be safe to use for osteopetrosis patients without inducing any significant untoward effects.

- How were 26 proteins in Table 2 selected? Table 2 should provide Fold changes of different comparisons (OP-Rx/OP-NoRx, OP-Rx/Ctrl, etc.).

Response: We conducted literature review of the differentially expressed proteins (with P value less than 0.05 and at least more than 2-fold change in their expression values). The review showed that 26 of these proteins have been individually reported to be implicated in osteogenesis. We consider that as an indirect support of some of the findings in this study. The pair wise fold changes as requested are indicated for all differentially expressed proteins in Supplementary table 2.

  1. Did fluconazole affect viability of treated cells?

Response: Examination under the microscope showed that more than 90% of the cells were viable and healthy prior to processing for proteomics analysis by mass spectrometry.

  1. Legends to figures should be clearer and more detailed.

Response: We have added more information for clarity of figure legends

  1. Figure 3 shows protein-protein interactions of “differentially expressed molecules between fluconazole treated and control cells”. What does it mean “control cells”? Healthy samples or untreated OP samples?

Response: The labeled control cells are samples from healthy subjects, the three samples groups have now been clearly defined in the figure legends.

Changes induced by fluconazole in OP samples (treated OP samples vs untreated OP samples) should also be shown. How were 251 proteins selected? – it is not clear. Please, describe Figure 3 in more detail, what do arrows of different color represent?

Response: We identified over 1500 unique proteins from the three sample groups of extracted fluconazole treated and untreated cells from dental pulp of osteopetrosis patients as well as healthy subjects. We then filtered through the 1500 proteins and selected only proteins that are differentially expressed statistically with P value less than 0.05 and at least more than 2-fold change in their expression values. The 251 proteins were then subjected to ingenuity pathway analysis as detailed under materials and methods. As the analysis was conducted for all the three sample groups in one single analysis matched set, the fold change difference was calculated for the highest mean condition in one of the three groups and the least mean condition in another group and third group is in between. The interactions of the proteins are as indicated in figure 3. The solid lines indicate direct interaction, and broken lines indicates indirect interaction. Details of the description are now indicated in the figure legend for clarity of reading. We have now provided the list of the 251 proteins as supplementary table 1.

- Please, specify the treatment regime with fluconazole in detail. In M&M, 3 concentrations are mentioned. Where was fluconazole obtained/purchased?

Response: Four concentrations (1µM, 2.5µM, 5µM, and 10µM were used. Pair wise of each of the treated does of OP was compared with un-treated OP samples. Only 250 that differed significantly in their expressions across all the four doses were then evaluated across the three sample groups of HC, OP not treated and Op treated samples. We have used10µM in the figure as symbolic representation of maximum dose used. We have clarified and added this statement under materials and methods section.

We used the commercially available fluconazole (Pfizer).

  1. Discussion is rather difficult to read, it should be more structured, the effect of CA activators in osteogenesis, including known data on fluconazole, should be discussed in more extent. The results should also be discussed in the context of the authors´ recent article on proteomic profiling of the dental pulp MSC (Alkhayal, Z.; Shinwari, Z.; Gaafar, A.; Alaiya, A. Proteomic Profiling of the First Human Dental Pulp Mesenchymal Stem/Stromal Cells from Carbonic Anhydrase II Deficiency Osteopetrosis Patients. Int. J. Mol. Sci. 2021, 22, 380).

Response: We have restructured the discussion section for clarity of reading and additional statements added to the discussion as suggested by the reviewer. We have also added into context our referenced previously published paper as appropriate.

  1. Minor comments:

-12.1 illustrated workflow is referenced as Figure 6, but it is Figure 7

Response: We strongly apologize for the incorrect figure citation and label. This was inadvertently due rearrangements of figures in an attempt to follow the journal template layout.

-12.2 in Table 2, line dealing with TGBFI, OP-Rx is given the both highest mean sample and lowest mean sample.

Response: We thank the reviewer for the meticulous reading and review of our manuscript and highlighting areas to improve the manuscript and we have corrected the deficiency as highlighted.

.

12.3 the sample names used in the figures should be explained in the figure legends

Response: All sample names has now been appropriately defined in the figure legends.

12.4 no reference to Figure 6 is given in the text of the article.

Response: Appropriate reference has been cited for figure 6.

Author's Reply to the Review Report (Reviewer 2)

Reviewer 2:

  1. As seen in the references, authors have performed similar analysis using Dental Pulp Mesenchymal Stem/Stromal Cells from CA-II Deficient Osteopetrosis Patients. However, authors did not describe in detail about previous their work and its results in the introduction and discussion sections, although the study also analyzed proteomic profiling using same source of cells, dental pulp mesenchymal stem/stromal cells from CA-II deficient patient and healthy donors.  I believe that it should be discussed and compared in detail with the present work and its results: which results are different between previous and present analysis and what are additionally acquired knowledge by adding Fluconazole-treatment. This is very essential part for discussion which the present manuscript has omitted.

Responses: A paragraph summarizing and discussing our previous publication has been added.

  1. In the Title and Abstract, there are unproved, speculating remarks.

In the Title, the manuscript titled as “A new use of azole for treatment of Osteopetrosis patients”. However, this study is not designed as use of azole for treatment of Osteopetrosis patients. This title may potentially mislead the readers. In the Abstract, authors also mentioned as “These findings propose that fluconazole might be a potential treatment agent for CAII- deficient OP patients.” I believe that this statement seems to be beyond the scope and approach of the present study and presented results. In addition, in conclusion section, we can find the similar overestimated remark.

Responses: The title has been modified to read, “Fluconazole induced protein changes in osteogenic and immune metabolic pathways of dental pulp MSCs' of Osteopetrosis patients”.

  1. Authors should emphasize and/or explain more in detail;

3.1-Why Fluconazole has been specifically chosen as a potential treatment agent among CA-activators.

Responses: Fluconazole, an antifungal medication, has been investigated as a potential treatment agent among carbonic anhydrase activators due to its ability to increase the activity of specific carbonic anhydrase (CA) isoforms. Here are a few of the reasons fluconazole was chosen for this study:

  1. Reuse of existing drugs: Fluconazole is a well-known antifungal medication that is widely used in clinical practice. Researchers frequently investigate the feasibility of repurposing existing drugs for new therapeutic applications, which can save time and money in drug development.
  2. Synergistic effects leading to improved therapeutic outcomes.
  3. Recent research, however, has looked into the potential use of certain azole derivatives as CA activators. These derivatives have specific modifications that allow them to activate CA rather than inhibit it. The rationale is to create new therapeutic approaches to target diseases where activating this enzyme may be beneficial, such as cancer, osteoporosis, and neurodegenerative disorders.

3.2) why do authors think that CA-activators has not been applied so far in clinical trials.

Response/Comment: It is important to note that this field of study is still in its early stages, and the clinical use of azoles as CA activators is not yet well established. Before obtaining regulatory approval, initiating clinical trials, it is critical to complete all basic and animal model research, including study design and protocol, objectives, eligibility criteria, treatment plan, and end-point of the study. We have indicated in the introduction that despite much being done CA inhibitor, in contrast, very few studies are available regarding CA activators.

3.3) for what authors aimed in the quantitative proteomics analysis and which kind of clinically relevant information could be obtained through the present analysis?

Response/comment:

As stated in the introduction, that analytical proteomics potentially powerful analytical technique for discovery of biomarker proteins for a variety of diseases. However, a small numbers of these studies have been translated into clinical use. Our analysis of protein profiles of azole-treated and non-treated osteopetrosis cells, identified treatment induced proteins. This information can provide insights into disease mechanisms, identify potential biomarkers, and discover therapeutic targets associated with specific monitoring treatment response, and predicting treatment outcomes. Quantitative proteomics provides a comprehensive and unbiased approach to biomarker discovery because it can capture dynamic changes in protein expression across the proteome. Additionally, it provides a global view of protein expression and regulation within a biological system. This information can aid in the development of targeted therapies and personalized medicine approaches. Overall, quantitative proteomics studies are critical for advancing our understanding of complex biological processes, disease mechanisms, and drug development efforts. It provide global view of protein expressions and regulations, and allows identifying important biological markers, pathways, and therapeutic targets.

3.4-Reviewers’ comment: Based on the results, what can be expected as next step trials to move forward toward a potential treatment agent for CAII- deficient OP patients as authors mentioned as a conclusive remark.

Response: Findings of this study is a proof of concept that fluconazole induced cellular protein changes related to pathogenesis of osteopetrosis. Therefore, our next step will be to look into the effect of Fluconazole in an osteopetrosis animal model. This will lead us to initiate clinical trials, looking at safety, and efficacy, in a controlled and regulated environment.

3.5- Reviewers’ comment: what is limitation/weakness of the present study to be further investigated.

Response: While basic research has limitations, it contributes to scientific advancement and innovation by generating new knowledge, understanding fundamental principles, and laying the groundwork for applied and translational research. The main limitation of this study is that Fluconazole was not tested in an osteopetrosis animal model.

3.6 Reviewers’ comment: what is advantage and meaningfulness using dental pulp mesenchymal stem/stromal cells for osteopetrosis-related protein analysis; how authors interpret the results (distinct 251 differentially expressed proteins or 26 closely associated with osteogenesis and osteopetrosis disease) out of dental pulp mesenchymal stem/stromal cells.

Response: Dental pulp mesenchymal stem cells (DP-MSCs) offer several benefits for research direct cellular therapy or tissue re-engineering. : Some of the osteopetrosis patients might benefit from cellular therapy or tissue re-engineering. Other rationale for working with DPMS include but not limited to the followings:

  1. Accessibility, Immunomodulatory Properties and low immunogenicity: DP-MSCs can be harvested from extracted or deciduous teeth without invasive procedures. DP-MSCs are easy to collect, useful for research or cellular banking for potential future autologous therapeutic use.
  2. Abundant Supply: Dental pulp contains a rich source of mesenchymal stem cells, allowing for the extraction of a significant number of DP-MSCs from a single tooth. This abundance provides researchers with a sufficient and renewable source of stem cells for various experiments, studies, and potential therapeutic applications.
  3. High Proliferative Potential and Multilineage differentiation: DP-MSCs multiply rapidly in culture without losing stem cell properties. DP-MSCs can be differentiated into osteoblasts, chondrocytes, adipocytes, and neural cells, which permits DP-MSCs to be good therapeutic targets.

The idea of using global proteomics analysis is that it offers unbiased analysis of complex protein mixtures and unravel proteins that are specific or related to disease being studied. We identified over 1500 unique proteins from extracted fluconazole treated and untreated cells from dental pulp. We then filtered through the 1500 proteins and selected only proteins that are differentially expressed statistically with P value less than 0.05 and at least more than 2-fold change in their expression values. A review of scientific literature showed that 26 of the 251 proteins have been individually reported to be implicated in osteogenesis. This might be an indirect was of validation of some of the observed findings in this study.

The idea of global untargeted protein analysis is in unbiased approach for disease-specific/related protein changes. By narrowing down the more than 1500 identified protein to only 250 that are significantly different due to treatment effect. Makes them easier to filter to a handful of biomarkers related to the disease being studied such as the 26 osteogenesis related protein as highlighted in this study.

Overall, in my opinion, many essential questions and discussions in the manuscript seem to be insufficiently described. I believe the manuscript has to be considerably revised for publication.

Response: All reviewers’ concerns have been appropriately responded as detailed above.

Reviewer 2 Report

Alkhayal et al. performed quantitative proteomics analysis on Fluconazole-treated dental pulp mesenchymal stem/stromal cells from CA-II deficient osteopetrosis patients.  The results seem to be interesting; however, the manuscript should add some more discussion about the analysis results supporting the conclusive remarks proposed in the abstracts and conclusion.

1.  As seen in the references, authors have performed similar analysis using Dental Pulp Mesenchymal Stem/Stromal Cells from CA-II Deficient Osteopetrosis Patients. However, authors did not describe in detail about previous their work and its results in the introduction and discussion sections, although the study also analyzed proteomic profiling using same source of cells,  dental pulp mesenchymal stem/stromal cells from CA-II deficient patient and healthy donors.  I believe that it should be discussed and compared in detail with the present work and its results: which results are different between previous and present analysis and what are additionally acquired knowledge by adding Fluconazole-treatment. This is very essential part for discussion which the present manuscript has omitted.

2. In the Title and Abstract, there are unproved, speculating remarks.

In the Title, the manuscript titled as “ A new use of azole for treatment of Osteopetrosis patients”. However, this study is not designed as use of azole for treatment of Osteopetrosis patients. This title may potentially mislead the readers. In the Abstract, authors also mentioned as “These findings propose that fluconazole might be a potential treatment agent for CAII- deficient OP patients.”. I believe that this statement seems to be beyond the scope and approach of the present study and presented results. Also in conclusion section, we can find the similar overestimated remark.

3.  Authors should emphasize and/or explain more in detail;

1) why Fluconazole has been specifically chosen as a potential treatment agent among CA-activators.

2) why do authors think that CA-activators has not been applied so far in clinical trials.

3) for what authors aimed in the quantitative proteomics analysis and which kind of clinically-relevant information could be obtained through the present analysis

4) Based on the results, what can be expected as next step trials to move forward toward a potential treatment agent for CAII- deficient OP patients as authors mentioned as a conclusive remark.

5) what is limitation/weakness of the present study to be further investigated.

6) what is advantage and meaningfulness using dental pulp mesenchymal stem/stromal cells for osteopetrosis-related protein analysis; how authors interpret the results (distinct 251 differentially expressed proteins or 26 closely associated with osteogenesis and osteopetrosis disease) out of dental pulp mesenchymal stem/stromal cells.

Overall, in my opinion, many essential questions and discussions in the manuscript seem to be insufficiently described. I believe the manuscript has to be considerably revised for publication.

Author Response

Author's Reply to the Review Report (Reviewer 2)

Reviewer 2:

  1. As seen in the references, authors have performed similar analysis using Dental Pulp Mesenchymal Stem/Stromal Cells from CA-II Deficient Osteopetrosis Patients. However, authors did not describe in detail about previous their work and its results in the introduction and discussion sections, although the study also analyzed proteomic profiling using same source of cells, dental pulp mesenchymal stem/stromal cells from CA-II deficient patient and healthy donors.  I believe that it should be discussed and compared in detail with the present work and its results: which results are different between previous and present analysis and what are additionally acquired knowledge by adding Fluconazole-treatment. This is very essential part for discussion which the present manuscript has omitted.

Responses: A paragraph summarizing and discussing our previous publication has been added.

  1. In the Title and Abstract, there are unproved, speculating remarks.

In the Title, the manuscript titled as “A new use of azole for treatment of Osteopetrosis patients”. However, this study is not designed as use of azole for treatment of Osteopetrosis patients. This title may potentially mislead the readers. In the Abstract, authors also mentioned as “These findings propose that fluconazole might be a potential treatment agent for CAII- deficient OP patients.” I believe that this statement seems to be beyond the scope and approach of the present study and presented results. In addition, in conclusion section, we can find the similar overestimated remark.

Responses: The title has been modified to read, “Fluconazole induced protein changes in osteogenic and immune metabolic pathways of dental pulp MSCs' of Osteopetrosis patients”.

  1. Authors should emphasize and/or explain more in detail;

3.1-Why Fluconazole has been specifically chosen as a potential treatment agent among CA-activators.

Responses: Fluconazole, an antifungal medication, has been investigated as a potential treatment agent among carbonic anhydrase activators due to its ability to increase the activity of specific carbonic anhydrase (CA) isoforms. Here are a few of the reasons fluconazole was chosen for this study:

  1. Reuse of existing drugs: Fluconazole is a well-known antifungal medication that is widely used in clinical practice. Researchers frequently investigate the feasibility of repurposing existing drugs for new therapeutic applications, which can save time and money in drug development.
  2. Synergistic effects leading to improved therapeutic outcomes.
  3. Recent research, however, has looked into the potential use of certain azole derivatives as CA activators. These derivatives have specific modifications that allow them to activate CA rather than inhibit it. The rationale is to create new therapeutic approaches to target diseases where activating this enzyme may be beneficial, such as cancer, osteoporosis, and neurodegenerative disorders.

3.2) why do authors think that CA-activators has not been applied so far in clinical trials.

Response/Comment: It is important to note that this field of study is still in its early stages, and the clinical use of azoles as CA activators is not yet well established. Before obtaining regulatory approval, initiating clinical trials, it is critical to complete all basic and animal model research, including study design and protocol, objectives, eligibility criteria, treatment plan, and end-point of the study. We have indicated in the introduction that despite much being done CA inhibitor, in contrast, very few studies are available regarding CA activators.

3.3) for what authors aimed in the quantitative proteomics analysis and which kind of clinically relevant information could be obtained through the present analysis?

Response/comment:

As stated in the introduction, that analytical proteomics potentially powerful analytical technique for discovery of biomarker proteins for a variety of diseases. However, a small numbers of these studies have been translated into clinical use. Our analysis of protein profiles of azole-treated and non-treated osteopetrosis cells, identified treatment induced proteins. This information can provide insights into disease mechanisms, identify potential biomarkers, and discover therapeutic targets associated with specific monitoring treatment response, and predicting treatment outcomes. Quantitative proteomics provides a comprehensive and unbiased approach to biomarker discovery because it can capture dynamic changes in protein expression across the proteome. Additionally, it provides a global view of protein expression and regulation within a biological system. This information can aid in the development of targeted therapies and personalized medicine approaches. Overall, quantitative proteomics studies are critical for advancing our understanding of complex biological processes, disease mechanisms, and drug development efforts. It provide global view of protein expressions and regulations, and allows identifying important biological markers, pathways, and therapeutic targets.

3.4-Reviewers’ comment: Based on the results, what can be expected as next step trials to move forward toward a potential treatment agent for CAII- deficient OP patients as authors mentioned as a conclusive remark.

Response: Findings of this study is a proof of concept that fluconazole induced cellular protein changes related to pathogenesis of osteopetrosis. Therefore, our next step will be to look into the effect of Fluconazole in an osteopetrosis animal model. This will lead us to initiate clinical trials, looking at safety, and efficacy, in a controlled and regulated environment.

3.5- Reviewers’ comment: what is limitation/weakness of the present study to be further investigated.

Response: While basic research has limitations, it contributes to scientific advancement and innovation by generating new knowledge, understanding fundamental principles, and laying the groundwork for applied and translational research. The main limitation of this study is that Fluconazole was not tested in an osteopetrosis animal model.

3.6 Reviewers’ comment: what is advantage and meaningfulness using dental pulp mesenchymal stem/stromal cells for osteopetrosis-related protein analysis; how authors interpret the results (distinct 251 differentially expressed proteins or 26 closely associated with osteogenesis and osteopetrosis disease) out of dental pulp mesenchymal stem/stromal cells.

Response: Dental pulp mesenchymal stem cells (DP-MSCs) offer several benefits for research direct cellular therapy or tissue re-engineering. : Some of the osteopetrosis patients might benefit from cellular therapy or tissue re-engineering. Other rationale for working with DPMS include but not limited to the followings:

  1. Accessibility, Immunomodulatory Properties and low immunogenicity: DP-MSCs can be harvested from extracted or deciduous teeth without invasive procedures. DP-MSCs are easy to collect, useful for research or cellular banking for potential future autologous therapeutic use.
  2. Abundant Supply: Dental pulp contains a rich source of mesenchymal stem cells, allowing for the extraction of a significant number of DP-MSCs from a single tooth. This abundance provides researchers with a sufficient and renewable source of stem cells for various experiments, studies, and potential therapeutic applications.
  3. High Proliferative Potential and Multilineage differentiation: DP-MSCs multiply rapidly in culture without losing stem cell properties. DP-MSCs can be differentiated into osteoblasts, chondrocytes, adipocytes, and neural cells, which permits DP-MSCs to be good therapeutic targets.

The idea of using global proteomics analysis is that it offers unbiased analysis of complex protein mixtures and unravel proteins that are specific or related to disease being studied. We identified over 1500 unique proteins from extracted fluconazole treated and untreated cells from dental pulp. We then filtered through the 1500 proteins and selected only proteins that are differentially expressed statistically with P value less than 0.05 and at least more than 2-fold change in their expression values. A review of scientific literature showed that 26 of the 251 proteins have been individually reported to be implicated in osteogenesis. This might be an indirect was of validation of some of the observed findings in this study.

The idea of global untargeted protein analysis is in unbiased approach for disease-specific/related protein changes. By narrowing down the more than 1500 identified protein to only 250 that are significantly different due to treatment effect. Makes them easier to filter to a handful of biomarkers related to the disease being studied such as the 26 osteogenesis related protein as highlighted in this study.

Overall, in my opinion, many essential questions and discussions in the manuscript seem to be insufficiently described. I believe the manuscript has to be considerably revised for publication.

Response: All reviewers’ concerns have been app

Round 2

Reviewer 1 Report

The manuscript was improved after revisions.

Author Response

N/A

Reviewer 2 Report

The revised manuscript has been improved. However, authors did not respond all suggestions and questionaire of the reviewer.

1 As seen in the references, authors have performed similar analysis using Dental Pulp Mesenchymal Stem/Stromal Cells from CA-II Deficient Osteopetrosis Patients. However, authors did not describe in detail about previous their work and its results in the introduction and discussion sections, although the study also analyzed proteomic profiling using same source of cells, dental pulp mesenchymal stem/stromal cells from CA-II deficient patient and healthy donors. I believe that it should be discussed and compared in detail with the present work and its results: which results are different between previous and present analysis and what are additionally acquired knowledge by adding Fluconazole-treatment. This is very essential part for discussion which the present manuscript has omitted.

Authors replied as:

Responses: A paragraph summarizing and discussing our previous publication has been added.

But I cannot find the detailed comparison and discussion in Introduction and Discussion sections  about previous authors´ results of proteomic analysis using almost same source of sample and  study design. I believe that this is very essential part for discussion which the present manuscript has omitted.

  1. Authors should emphasize and/or explain more in detail;

3.1) Why Fluconazole has been specifically chosen as a potential treatment agent among CA-activators.

3.3) for what authors aimed in the quantitative proteomics analysis and which kind of clinically relevant information could be obtained through the present analysis?

The explanations in authors reply are only confined to too general descriptions. Further, it has not been added properly in the revised manuscript. 

Author Response

ID: ijms-2500580

We thank the reviewers and the Editorial Office for their second round critical review of our manuscript. We are pleased to inform you that the manuscript has been thoroughly revised and all the raised concerns were responded to adequately. Please find below our point-by-point responses to all the raised questions/ suggestions in blue color
